# CROSSFORMER: A VERSATILE VISION TRANSFORMER HINGING ON CROSS-SCALE ATTENTION

**Wenxiao Wang[1,2], Lu Yao[1], Long Chen[3], Binbin Lin[4,∗], Deng Cai[1], Xiaofei He[1] & Wei Liu[2,∗]**
[1]State Key Lab of CAD & CG, Zhejiang University
[2]Data Platform, Tencent
[3]Columbia University
[4]School of Software Technology, Zhejiang University

## ABSTRACT

Transformers have made great progress in dealing with computer vision tasks. However, existing vision transformers have not yet possessed the ability of building the interactions among features of different scales, which is perceptually important to visual inputs. The reasons are two-fold: (1) Input embeddings of each layer are equal-scale, so no cross-scale feature can be extracted; (2) to lower the computational cost, some vision transformers merge adjacent embeddings inside the self-attention module, thus sacrificing small-scale (fine-grained) features of the embeddings and also disabling the cross-scale interactions. To this end, we propose **C**ross-scale **E**mbedding **L**ayer (CEL) and **L**ong **S**hort **D**istance **A**ttention (LSDA). On the one hand, CEL blends each embedding with multiple patches of different scales, providing the self-attention module itself with cross-scale features. On the other hand, LSDA splits the self-attention module into a short-distance one and a long-distance counterpart, which not only reduces the computational burden but also keeps both small-scale and large-scale features in the embeddings. Through the above two designs, we achieve cross-scale attention. Besides, we put forward a dynamic position bias for vision transformers to make the popular relative position bias apply to variable-sized images. Hinging on the cross-scale attention module, we construct a versatile vision architecture, dubbed CrossFormer, which accommodates variable-sized inputs. Extensive experiments show that CrossFormer outperforms the other vision transformers on image classification, object detection, instance segmentation, and semantic segmentation tasks. [1]

## 1 INTRODUCTION

It turns out that transformer (Vaswani et al., 2017; Devlin et al., 2019; Brown et al., 2020) has achieved great success in the field of natural language processing (NLP). Benefitting from its self-attention module, transformer is born with the key ability to build long-distance dependencies. Since long-distance dependencies are also needed by a number of vision tasks (Zhang & Yang, 2021; Chu et al., 2021), a surge of research work (Dosovitskiy et al., 2021; Touvron et al., 2021; Wang et al., 2021) has been conducted to explore various transformer-based vision architectures.

A transformer requires a sequence of embeddings[2](*e.g.*, word embeddings) as input. To adapt this requirement to typical vision tasks, most existing vision transformers (Dosovitskiy et al., 2021; Touvron et al., 2021; Wang et al., 2021; Liu et al., 2021b) produce embeddings by splitting an input image into equal-sized patches. For example, a $224 \times 224$ image can be split into $56 \times 56$ patches of size $4 \times 4$, and these patches are projected through a linear layer to yield an embedding sequence. Inside a certain transformer, self-attention is engaged to build the interactions between any two embeddings. Thus, the computational or memory cost of the self-attention module is $O(N^2)$, where $N$ is the length of an embedding sequence. Such a cost is too big for a visual input because its embedding sequence is much longer than that of NLP. Therefore, the recently proposed vision

---

∗The corresponding authors.

[1]The code has been released: https://github.com/cheerss/CrossFormer
[2]In this paper, we also use "embeddings" to represent the input of each layer.

transformers (Wang et al., 2021; Liu et al., 2021b; Lin et al., 2021) develop multiple substitutes to approximate the vanilla self-attention module with a lower cost.

Though the aforementioned vision transformers have made some progress, they do not explicitly utilize features of different scales, whereas multi-scale features are very vital for a lot of vision tasks. For example, an image often contains many objects of different scales, and to fully understand the image, building the interactions among those objects is helpful. Besides, some particular tasks such as instance segmentation need the interactions between large-scale (coarse-grained) features and small-scale (fine-grained) ones. Existing vision transformers fail to deal with the above cases due to two reasons: (1) The embeddings are generated from equal-sized patches. Though these patches theoretically have a chance to extract any scale features if only the receptive field is large enough, it is difficult to promise that they can learn appropriate multi-scale features automatically in practice. (2) Inside the self-attention module, adjacent embeddings are often grouped together and merged (Wang et al., 2021). Since the number of groups is smaller than that of embeddings, such a behavior can reduce the computational budget of the self-attention. In this case, however, even if embeddings have both small-scale and large-scale features, merging operations will lose the small-scale features of each individual embedding, thereby disabling the cross-scale attention.

To enable the building of cross-scale interactions, we co-design a novel embedding layer and a self-attention module as follows. 1) *Cross-scale Embedding Layer (CEL)* – Following Wang et al. (2021), we also employ a pyramid structure for our transformer, which naturally splits the vision transformer model into multiple stages. CEL appears at the start of each stage, which receives last stage's output (or an input image) as input and samples patches with multiple kernels of different scales (*e.g.*, $4 \times 4$ or $8 \times 8$). Then, each embedding is constructed by projecting and concatenating these patches. Through this way, we enforce some dimensions (*e.g.*, dimensions from $4 \times 4$ patches) to focus on small-scale features only, while others (*e.g.*, those from $8 \times 8$ patches) have a chance to learn large-scale features, leading to an embedding with explicitly cross-scale features. 2) *Long Short Distance Attention (LSDA)* – We propose a substitute of the vanilla self-attention, but to preserve small-scale features, the embeddings will not be merged. In contrast, we split the self-attention module into *Short Distance Attention* (SDA) and *Long Distance Attention* (LDA). SDA builds the dependencies among neighboring embeddings, while LDA takes charge of the dependencies among embeddings far away from each other. The proposed LSDA can also reduce the cost of the self-attention module like previous studies (Wang et al., 2021), but different from them, LSDA does not undermine either small-scale or large-scale features. As a result, attention with cross-scale interactions is enabled.

Besides, following prior work (Shaw et al., 2018; Liu et al., 2021b), we employ a relative position bias for embeddings' position representations. The Relative Position Bias (RPB) only supports fixed image/group size[3]. However, image size for many vision tasks such as object detection is variable, so does group size for many architectures, including ours. To make the RPB more flexible, we further introduce a trainable module called *Dynamic Position Bias* (DPB), which receives two embeddings' relative distance as input and outputs their position bias. The DPB module is optimized end-to-end in the training phase, inducing an ignorable cost but making RPB apply to variable image/group size.

All our proposed modules can be implemented with about ten lines of code. Based on them, we construct four versatile vision transformers of different sizes, dubbed *CrossFormers*. Other than image classification, the proposed CrossFormer can handle a variety of tasks with variable-sized inputs such as object detection. Experiments on four representative vision tasks (*i.e.*, image classification, object detection, instance segmentation, and semantic segmentation) demonstrate that CrossFormer outperforms the other state-of-the-art vision transformers on all the tasks. Remarkably, the performance gains brought by CrossFormer are substantially significant on dense prediction tasks, *e.g.*, object detection and instance/semantic segmentation.

It is worth highlighting our contributions as follows:

- We propose cross-scale embedding layer (CEL) and long short distance attention (LSDA), which together compensate for existing transformers' incapability of building cross-scale attention.

- The dynamic position bias module (DPB) is further proposed to make the relative position bias more flexible, *i.e.*, accommodating variable image size or group size.

---

[3]Some vision transformers split input embeddings into several groups. Group size means the number of embeddings in a group.

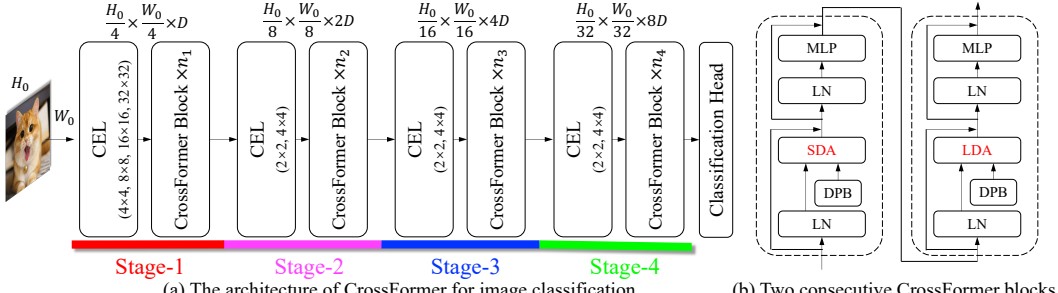

(a) The architecture of CrossFormer for image classification.

(b) Two consecutive CrossFormer blocks.

Figure 1: (a) The architecture of CrossFormer for classification. The input size is $H_0 \times W_0$, and the size of feature maps in each stage is shown on the top. *Stage-i* consists of a CEL and $n_i$ CrossFormer blocks. Numbers in CELs represent kernels' sizes used for sampling patches. (b) The inner structure of two consecutive CrossFormer blocks. SDA and LDA appear alternately in different blocks.

- Multiple CrossFormers with different sizes are constructed, and we corroborate their effectiveness through sufficient experiments on four representative vision tasks.

## 2 BACKGROUND

**Vision Transformers.** Motivated by the transformers developed for NLP, researchers design specific visual transformers for vision tasks to take full advantage of their powerful attention mechanism. In particular, ViT and DeiT transfer the original transformer Vaswani et al. (2017) to vision tasks (Touvron et al., 2021; Dosovitskiy et al., 2021), achieving impressive performance. Later, PVT (Wang et al., 2021), HVT (Pan et al., 2021), Swin (Liu et al., 2021b), and ViTAE (Xu et al., 2021b) introduce a pyramid structure into the visual transformers, greatly decreasing the number of patches in the later layers of a respective model. They also extend the visual transformers to other vision tasks like object detection and segmentation (Wang et al., 2021; Liu et al., 2021b).

**Cross-scale Feature Extraction.** Szegedy et al. (2015); Tan & Le (2019) use multi-scale convolutional kernels at every layer of models for cross-scale features, while the largest kernel size can only be $7 \times 7$, and larger kernel size will induce unaccpetable computational budget. For vision transformers, CoaT (Xu et al., 2021a) uses multi-scale features at later layers of models by mixing features from different layers; CViT (Chen et al., 2021a) keeps embeddings of different scales in different branches, and introduces cross-scale interaction through self-attention between branches. More discussions can be seen in the appendix C.1.

**Sparse Self-attention.** Many substitutes have been proposed (Liu et al., 2021b; Wang et al., 2021; Chu et al., 2021; Child et al., 2019; Beltagy et al., 2020; Zaheer et al., 2020) to alleviate the cost of the vanilla self-attention module. Instead of going self-attention among all embeddings, each embedding only interacts with part of other embeddings, thus changing dense self-attention to sparse attention. More discussions can be seen in the appendix C.2.

**Position Representations.** To make the respective model aware of position information of embeddings, many different position representations (Vaswani et al., 2017) are proposed. For example, Dosovitskiy et al. (2021) directly added the embeddings with the vectors that contain absolute position information. In contrast, Relative Position Bias (RPB) (Shaw et al., 2018) resorts to position information indicating the relative distance of two embeddings.

## 3 CROSSFORMER

The overall architecture of CrossFormer is plotted in Figure 1. Following (Wang et al., 2021; Liu et al., 2021b; Lin et al., 2021), CrossFormer also employs a pyramid structure, which naturally splits the transformer model into four stages. Each stage consists of a cross-scale embedding layer (CEL, Section 3.1) and several CrossFormer blocks (Section 3.2). A CEL receives last stage's output (or an input image) as input and generates cross-scale embeddings. In this process, CEL (except that in *Stage-1*) reduces the number of embeddings to a quarter while doubles their dimensions for a pyramid structure. Then, several CrossFormer blocks, each of which involves long short distance attention (LSDA) and dynamic position bias (DPB), are set up after CEL. A specialized head (*e.g.*, the classification head in Figure 1) follows after the final stage accounting for a specific task.

## 3.1 CROSS-SCALE EMBEDDING LAYER (CEL)

Cross-scale embedding layer (CEL) is leveraged to generate input embeddings for each stage. Figure 2 takes the first CEL, which is ahead of *Stage-1*, as an example. It receives an image as input, then sampling patches using four kernels of different sizes. The stride of four kernels is kept the same so that they generate the same number of embeddings[4]. As we can observe in Figure 2, every four corresponding patches have the same center but different scales, and all these four patches will be projected and concatenated as one embedding. In practice, the process of sampling and projecting can be fulfilled through four convolutional layers.

For a cross-scale embedding, one problem is how to set the projected dimension of each scale. The computational budget of a convolutional layer is proportional to $K^2D^2$, where $K$ and $D$ represent kernel size and input/output dimension, respectively (assuming the input dimension equals to the output dimension). Therefore, given the same dimension, a large kernel consumes more budget than a smaller one. To control the total budget of the CEL, we use a lower dimension for large kernels while a higher dimension for small kernels. Figure 2 provides the specific allocation rule in its subtable, and a $128$-dimensional example is given. Compared with allocating the dimension equally, our scheme saves much computational cost but does not explicitly affect the model's performance. The cross-scale embedding layers in other stages work in a similar way. As shown in Figure 1, CELs for *Stage-2/3/4* use two different kernels ($2 \times 2$ and $4 \times 4$). Further, to form a pyramid structure, the strides of CELs for *Stage-2/3/4* are set as $2 \times 2$ to reduce the number of embeddings to a quarter.

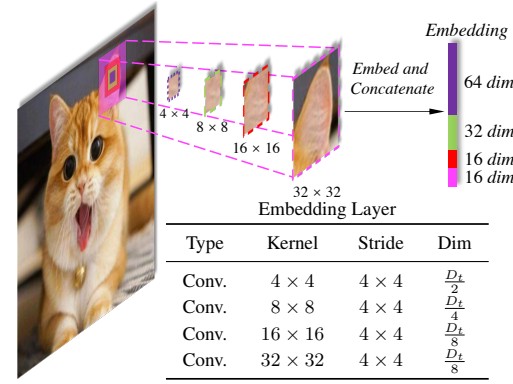

| Type | Kernel | Stride | Dim |
|------|--------|--------|-----|
| Conv. | $4 \times 4$ | $4 \times 4$ | $\frac{D_t}{2}$ |
| Conv. | $8 \times 8$ | $4 \times 4$ | $\frac{D_t}{4}$ |
| Conv. | $16 \times 16$ | $4 \times 4$ | $\frac{D_t}{8}$ |
| Conv. | $32 \times 32$ | $4 \times 4$ | $\frac{D_t}{8}$ |

Figure 2: Illustration of the CEL layer. The input image is sampled by four different kernels (*i.e.*, $4 \times 4, 8 \times 8, 16 \times 16, 32 \times 32$) with same stride $4 \times 4$. Each embedding is constructed by projecting and concatenating the four patches. $D_t$ means the total dimension of the embedding.

## 3.2 CROSSFORMER BLOCK

Each CrossFormer block consists of a long short distance attention module (*i.e.*, LSDA, which involves a short distance attention (SDA) module or a long distance attention (LDA) module) and a multilayer perceptron (MLP). As shown in Figure 1b, SDA and LDA appear alternately in different blocks, and the dynamic position bias (DPB) module works in both SDA and LDA for obtaining embeddings' position representations. Following the prior vision transformers, residual connections are used in each block.

### 3.2.1 LONG SHORT DISTANCE ATTENTION (LSDA)

We split the self-attention module into two parts: short distance attention (SDA) and long distance attention (LDA). For SDA, every $G \times G$ adjacent embeddings are grouped together. Figure 3a gives an example where $G = 3$. For LDA with input of size $S \times S$, the embeddings are sampled with a fixed interval $I$. For example in Figure 3b ($I = 3$), all embeddings with a red border belong to a group, and those with a yellow border comprise another group. The group's height or width for LDA is computed as $G = S/I$ (*i.e.*, $G = 3$ in this example). After grouping embeddings, both SDA and LDA employ the vanilla self-attention within each group. As a result, the memory/computational cost of the self-attention module is reduced from $O(S^4)$ to $O(S^2G^2)$, and $G \ll S$ in most cases.

It is worth noting that the effectiveness of LDA also benefits from cross-scale embeddings. Specifically, we draw all the patches comprising two embeddings in Figure 3b. As we can see, the small-scale patches of two embeddings are non-adjacent, so it is difficult to judge their relationship without the help of the context. In other words, it will be hard to build the dependency between these two embeddings if they are only constructed by small-scale patches (*i.e.*, single-scale feature). On the contrary, adjacent large-scale patches provide sufficient context to link these two embeddings, which makes long-distance cross-scale attention easier and more meaningful.

---

[4]The image will be padded if necessary.

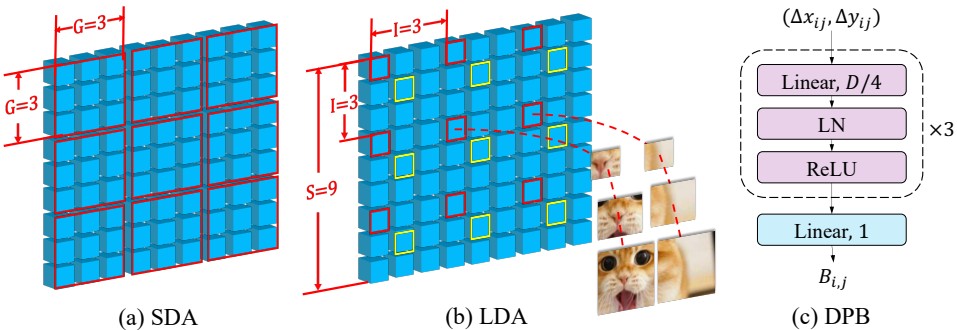

| (a) SDA | (b) LDA | (c) DPB |

Figure 3: (a) Short distance attention (SDA). Embeddings (blue cubes) are grouped by red boxes. (b) Long distance attention (LDA). Embeddings with the same color borders belong to the same group. Large patches of embeddings in the same group are adjacent. (c) Dynamic position bias (DBP). The dimensions of intermediate layers are $D/4$, and the output is a scalar.

We provide the pseudo-code of LSDA in the appendix (A.1). Based on the vanilla multi-head self-attention, LSDA can be implemented with only ten lines of code. Further, only *reshape* and *permute* operations are used, so no extra computational cost is introduced.

### 3.2.2 DYNAMIC POSITION BIAS (DPB)

Relative position bias (RPB) indicates embeddings' relative position by adding a bias to their attention. Formally, the LSDA's attention map with RPB becomes:

$$\texttt{Attention} = \texttt{Softmax}(\boldsymbol{Q}\boldsymbol{K}^T/\sqrt{d} + \boldsymbol{B})\boldsymbol{V}, \tag{1}$$

where $\boldsymbol{Q}, \boldsymbol{K}, \boldsymbol{V} \in \mathbb{R}^{G^2 \times D}$ represent *query, key, value* in the self-attention module, respectively. $\sqrt{d}$ is a constant normalizer, and $\boldsymbol{B} \in \mathbb{R}^{G^2 \times G^2}$ is the RPB matrix. In previous work (Liu et al., 2021b), $\boldsymbol{B}_{i,j} = \hat{\boldsymbol{B}}_{\Delta x_{ij}, \Delta y_{ij}}$, where $\hat{\boldsymbol{B}}$ is a fixed-sized matrix, and $(\Delta x_{ij}, \Delta y_{ij})$ is the coordinate distance between the $i_{th}$ and $j_{th}$ embeddings. It is obvious that the image/group size is restricted in case that $(\Delta x_{ij}, \Delta y_{ij})$ exceeds the size of $\hat{B}$. In contrast, we propose an MLP-based module called DPB to generate the relative position bias dynamically, *i.e.*,

$$\boldsymbol{B}_{i,j} = DPB(\Delta x_{ij}, \Delta y_{ij}). \tag{2}$$

The structure of DPB is displayed in Figure 3c. Its non-linear transformation consists of three fully-connected layers with layer normalization (Ba et al., 2016) and ReLU (Nair & Hinton, 2010). The input dimension of DPB is 2, *i.e.*, $(\Delta x_{ij}, \Delta y_{ij})$, and intermediate layers' dimension is set as $D/4$, where $D$ is the dimension of embeddings. The output $B_{ij}$ is a scalar, encoding the relative position feature between the $i_{th}$ and $j_{th}$ embeddings. DPB is a trainable module optimized along with the whole transformer model. It can deal with any image/group size without worrying about the bound of $(\Delta x_{ij}, \Delta y_{ij})$. In the appendix (A.2), we prove that DPB is equivalent to RPB if the image/group size is fixed. In this case, we can transform a trained DPB to RPB in the test phase. We also provide an efficient $O(G^2)$ implementation of DPB when image/group size is variable (the complexity is $O(G^4)$ in a normal case because $\boldsymbol{B} \in \mathbb{R}^{G^2 \times G^2}$).

### 3.3 VARIANTS OF CROSSFORMER

Table 1 lists the detailed configurations of CrossFormer's four variants (-T, -S, -B, and -L for tiny, small, base, and large, respectively) for image classification. To re-use the pre-trained weights, the models for other tasks (*e.g.*, object detection) employ the same backbones as classification except that they may use different $G$ and $I$. Specifically, besides the configurations same to classification, we also test with $G_1 = G_2 = 14, I_1 = 16$, and $I_2 = 8$ for the detection (and segmentation) models' first two stages to adapt to larger images. The specific configurations are described in the appendix (A.3). Notably, the group size or the interval (*i.e.*, $G$ or $I$) does not affect the shape of weight tensors, so the backbones pre-trained on ImageNet can be readily fine-tuned on other tasks even though they use different $G$ or $I$.

Table 1: Variants of CrossFormer for image classification. The example input size is $224 \times 224$. $S$ represents the feature maps' height (and width) of each stage. $D$ and $H$ mean embedding dimensions and the number of heads in the multi-head self-attention module, respectively. $G$ and $I$ are group size and interval for SDA and LDA, respectively.

| | Output Size | Layer Name | CrossFormer-T | CrossFormer-S | CrossFormer-B | CrossFormer-L |
|---|---|---|---|---|---|---|
| Stage-1 | $56 \times 56$ $(S_1 = 56)$ | Cross Embed. | Kernel size: $4 \times 4, 8 \times 8, 16 \times 16, 32 \times 32$, Stride=4 | | | |
| | | SDA/LDA MLP | $\begin{bmatrix} D_1 = 64 \\ H_1 = 2 \\ G_1 = 7 \\ I_1 = 8 \end{bmatrix} \times 1$ | $\begin{bmatrix} D_1 = 96 \\ H_1 = 3 \\ G_1 = 7 \\ I_1 = 8 \end{bmatrix} \times 2$ | $\begin{bmatrix} D_1 = 96 \\ H_1 = 3 \\ G_1 = 7 \\ I_1 = 8, \end{bmatrix} \times 2$ | $\begin{bmatrix} D_1 = 128 \\ H_1 = 4 \\ G_1 = 7 \\ I_1 = 8 \end{bmatrix} \times 2$ |
| Stage-2 | $28 \times 28$ $(S_2 = 28)$ | Cross Embed. | Kernel size: $2 \times 2, 4 \times 4$, Stride=2 | | | |
| | | SDA/LDA MLP | $\begin{bmatrix} D_2 = 128 \\ H_2 = 4 \\ G_2 = 7 \\ I_2 = 4 \end{bmatrix} \times 1$ | $\begin{bmatrix} D_2 = 192 \\ H_2 = 6 \\ G_2 = 7 \\ I_2 = 4 \end{bmatrix} \times 2$ | $\begin{bmatrix} D_2 = 192 \\ H_2 = 6 \\ G_2 = 7 \\ I_2 = 4 \end{bmatrix} \times 2$ | $\begin{bmatrix} D_2 = 256 \\ H_2 = 8 \\ G_2 = 7 \\ I_2 = 4 \end{bmatrix} \times 2$ |
| Stage-3 | $14 \times 14$ $(S_3 = 14)$ | Cross Embed. | Kernel size: $2 \times 2, 4 \times 4$, Stride=2 | | | |
| | | SDA/LDA MLP | $\begin{bmatrix} D_3 = 256 \\ H_3 = 8 \\ G_3 = 7 \\ I_3 = 2 \end{bmatrix} \times 8$ | $\begin{bmatrix} D_3 = 384 \\ H_3 = 12 \\ G_3 = 7 \\ I_3 = 2 \end{bmatrix} \times 6$ | $\begin{bmatrix} D_3 = 384 \\ H_3 = 12 \\ G_3 = 7 \\ I_3 = 2 \end{bmatrix} \times 18$ | $\begin{bmatrix} D_3 = 512 \\ H_3 = 16 \\ G_3 = 7 \\ I_3 = 2 \end{bmatrix} \times 18$ |
| Stage-4 | $7 \times 7$ $(S_4 = 7)$ | Cross Embed. | Kernel size: $2 \times 2, 4 \times 4$, Stride=2 | | | |
| | | SDA/LDA MLP | $\begin{bmatrix} D_4 = 512 \\ H_4 = 16 \\ G_4 = 7 \\ I_4 = 1 \end{bmatrix} \times 6$ | $\begin{bmatrix} D_4 = 768 \\ H_4 = 24 \\ G_4 = 7 \\ I_4 = 1 \end{bmatrix} \times 2$ | $\begin{bmatrix} D_4 = 768 \\ H_4 = 24 \\ G_4 = 7 \\ I_4 = 1 \end{bmatrix} \times 2$ | $\begin{bmatrix} D_4 = 1024 \\ H_4 = 32 \\ G_4 = 7 \\ I_4 = 1 \end{bmatrix} \times 2$ |
| Head | $1 \times 1$ | Avg Pooling | Kernel size: $7 \times 7$ | | | |
| | | Linear | Classes: 1000 | | | |

# 4 EXPERIMENTS

The experiments are carried out on four challenging tasks: image classification, object detection, instance segmentation, and semantic segmentation. To entail a fair comparison, we keep the same data augmentation and training settings as the other vision transformers as far as possible. The competitors are all competitive vision transformers, including DeiT (Touvron et al., 2021), PVT (Wang et al., 2021), T2T-ViT (Yuan et al., 2021), TNT (Han et al., 2021), CViT (Chen et al., 2021a), Twins (Chu et al., 2021), Swin (Liu et al., 2021b), NesT (Zhang et al., 2021b), CvT (Wu et al., 2021), ViL (Zhang et al., 2021a), CAT (Lin et al., 2021), ResT (Zhang & Yang, 2021), TransCNN (Liu et al., 2021a), Shuffle (Huang et al., 2021), BoTNet (Srinivas et al., 2021), and RegionViT (Chen et al., 2021b).

## 4.1 IMAGE CLASSIFICATION

**Experimental Settings.** The experiments on image classification are done with the ImageNet (Russakovsky et al., 2015) dataset. The models are trained on 1.28M training images and tested on 50K validation images. The same training settings as the other vision transformers are adopted. In particular, we use an AdamW (Kingma & Ba, 2015) optimizer training for 300 epochs with a cosine decay learning rate scheduler, and 20 epochs of linear warm-up are used. The batch size is 1,024 split on 8 V100 GPUs. An initial learning rate of 0.001 and a weight decay of 0.05 are used. Besides, we use drop path rate of $0.1, 0.2, 0.3, 0.5$ for CrossFormer-T, CrossFormer-S, CrossFormer-B, CrossFormer-L, respectively. Further, Similar to Swin (Liu et al., 2021b), RandAugment (Cubuk et al., 2020), Mixup (Zhang et al., 2018a), Cutmix (Yun et al., 2019), random erasing (Zhong et al., 2020), and stochastic depth (Huang et al., 2016) are used for data augmentation.

**Results.** The results are shown in Table 2. As we can see, CrossFormer achieves the highest accuracy with parameters and FLOPs comparable to other state-of-the-art vision transformer structures. In specific, compared against strong baselines DeiT, PVT, and Swin, our CrossFormer outperforms them at least absolute $1.2\%$ in accuracy on small models. Further, though RegionViT achieves the same accuracy ($82.5\%$) as ours on a small model, it is $0.7\%$ ($84.0\%$ vs. $83.3\%$) absolutely lower than ours on the large model.

## 4.2 OBJECT DETECTION AND INSTANCE SEGMENTATION

**Experimental Settings.** The experiments on object detection and instance segmentation are both done on the COCO 2017 dataset (Lin et al., 2014), which contains 118K training and 5K val images.

Table 2: Results on the ImageNet validation set. The input size is $224 \times 224$ for most models, while is $384 \times 384$ for the model with a $\dagger$. Results of other architectures are drawn from original papers.

| Architectures | #Params | FLOPs | Acc. |
|---|---|---|---|
| PVT-S | 24.5M | 3.8G | 79.8% |
| RegionViT-T | 13.8M | 2.4G | 80.4% |
| Twins-SVT-S | 24.0M | 2.8G | 81.3% |
| **CrossFormer-T** | 27.8M | 2.9G | **81.5%** |
| DeiT-S | 22.1M | 4.6G | 79.8% |
| T2T-ViT | 21.5M | 5.2G | 80.7% |
| CViT-S | 26.7M | 5.6G | 81.0% |
| PVT-M | 44.2M | 6.7G | 81.2% |
| TNT-S | 23.8M | 5.2G | 81.3% |
| Swin-T | 29.0M | 4.5G | 81.3% |
| NesT-T | 17.0M | 5.8G | 81.5% |
| CvT-13 | 20.0M | 4.5G | 81.6% |
| ResT | 30.2M | 4.3G | 81.6% |
| CAT-S | 37.0M | 5.9G | 81.8% |
| ViL-S | 24.6M | 4.9G | 81.8% |
| RegionViT-S | 30.6M | 5.3G | **82.5%** |
| **CrossFormer-S** | 30.7M | 4.9G | **82.5%** |

| Architectures | #Params | FLOPs | Acc. |
|---|---|---|---|
| BoTNet-S1-59 | 33.5M | 7.3G | 81.7% |
| PVT-L | 61.4M | 9.8G | 81.7% |
| CvT-21 | 32.0M | 7.1G | 82.5% |
| CAT-B | 52.0M | 8.9G | 82.8% |
| Swin-S | 50.0M | 8.7G | 83.0% |
| RegionViT-M | 41.2M | 7.4G | 83.1% |
| Twins-SVT-B | 56.0M | 8.3G | 83.1% |
| NesT-S | 38.0M | 10.4G | 83.3% |
| **CrossFormer-B** | 52.0M | 9.2G | **83.4%** |
| DeiT-B | 86.0M | 17.5G | 81.8% |
| DeiT-B$^\dagger$ | 86.0M | 55.4G | 83.1% |
| ViL-B | 55.7M | 13.4G | 83.2% |
| RegionViT-B | 72.0M | 13.3G | 83.3% |
| Twins-SVT-L | 99.2M | 14.8G | 83.3% |
| Swin-B | 88.0M | 15.4G | 83.3% |
| NesT-B | 68.0M | 17.9G | 83.8% |
| **CrossFormer-L** | 92.0M | 16.1G | **84.0%** |

Table 3: Object detection results on COCO 2017 *val* set with RetinaNets as detectors. Results for Swin are drawn from Twins as Swin does not report results on RetinaNet. Results in blue fonts are the second-placed ones. CrossFormers with $\ddagger$ use different group sizes from classification models. (More details are put in the appendix (A.3))

| Method | Backbone | #Params | FLOPs | $AP^b$ | $AP^b_{50}$ | $AP^b_{75}$ | $AP^b_S$ | $AP^b_M$ | $AP^b_L$ |
|---|---|---|---|---|---|---|---|---|---|
| | ResNet-50 | 37.7M | 234.0G | 36.3 | 55.3 | 38.6 | 19.3 | 40.0 | 48.8 |
| | CAT-B | 62.0M | 337.0G | 41.4 | 62.9 | 43.8 | 24.9 | 44.6 | 55.2 |
| | Swin-T | 38.5M | 245.0G | 41.5 | 62.1 | 44.2 | 25.1 | 44.9 | 55.5 |
| | PVT-M | 53.9M | − | 41.9 | 63.1 | 44.3 | 25.0 | 44.9 | 57.6 |
| | ViL-M | 50.8M | 338.9G | 42.9 | 64.0 | 45.4 | 27.0 | 46.1 | 57.2 |
| | RegionViT-B | 83.4M | 308.9G | 43.3 | 65.2 | 46.4 | 29.2 | 46.4 | 57.0 |
| | TransCNN-B | 36.5M | − | 43.4 | 64.2 | 46.5 | 27.0 | 47.4 | 56.7 |
| RetinaNet | **CrossFormer-S** | 40.8M | 282.0G | **44.4** (+1.0) | 65.8 | 47.4 | 28.2 | 48.4 | 59.4 |
| 1× schedule | **CrossFormer-S$^\ddagger$** | 40.8M | 272.1G | **44.2** (+0.8) | 65.7 | 47.2 | 28.0 | 48.0 | 59.1 |
| | ResNet101 | 56.7M | 315.0G | 38.5 | 57.8 | 41.2 | 21.4 | 42.6 | 51.1 |
| | PVT-L | 71.1M | 345.0G | 42.6 | 63.7 | 45.4 | 25.8 | 46.0 | 58.4 |
| | Twins-SVT-B | 67.0M | 322.0G | 44.4 | 66.7 | 48.1 | 28.5 | 48.9 | 60.6 |
| | RegionViT-B+ | 84.5M | 328.2G | 44.6 | 66.4 | 47.6 | 29.6 | 47.6 | 59.0 |
| | Swin-B | 98.4M | 477.0G | 44.7 | 65.9 | 49.2 | − | − | − |
| | Twins-SVT-L | 110.9M | 455.0G | 44.8 | 66.1 | 48.1 | 28.4 | 48.3 | 60.1 |
| | **CrossFormer-B** | 62.1M | 389.0G | **46.2** (+1.4) | 67.8 | 49.5 | 30.1 | 49.9 | 61.8 |
| | **CrossFormer-B$^\ddagger$** | 62.1M | 379.1G | **46.1** (+1.3) | 67.7 | 49.0 | 29.5 | 49.9 | 61.5 |

We use MMDetection-based (Chen et al., 2019) RetinaNet (Lin et al., 2020) and Mask R-CNN (He et al., 2017) as the object detection and instance segmentation head, respectively. For both tasks, the backbones are initialized with the weights pre-trained on ImageNet. Then the whole models are trained with batch size 16 on 8 V100 GPUs, and an AdamW optimizer with an initial learning rate of $1 \times 10^{-4}$ is used. Following previous works, we adopt 1× training schedule (*i.e.*, the models are trained for 12 epochs) when taking RetinaNets as detectors, and images are resized to 800 pixels for the short side. While for Mask R-CNN, both 1× and 3× training schedules are used. It is noted that multi-scale training (Carion et al., 2020) is also employed when taking 3× training schedules.

**Results.** The results on RetinaNet and Mask R-CNN are shown in Table 3 and Table 4, respectively. As we can see, the second-placed architecture changes along with the experiment, that is, these architectures may perform well on one task but poorly on another task. In contrast, our CrossFormer outperforms all the others on both tasks (detection and segmentation) with both model sizes (small and base). Further, CrossFormer's performance gain over the other architectures gets sharper when enlarging the model, indicating that CrossFormer enjoys greater potentials.

### 4.3 SEMANTIC SEGMENTATION

**Experimental Settings.** ADE20K (Zhou et al., 2017) is used as the benchmark for semantic segmentation. It covers a broad range of 150 semantic categories, including 20K images for training

Table 4: Object detection and instance segmentation results on COCO *val* 2017 with Mask R-CNNs as detectors. $AP^b$ and $AP^m$ are box average precision and mask average precision, respectively.

| Method | Backbone | #Params | FLOPs | $AP^b$ | $AP^b_{50}$ | $AP^b_{75}$ | $AP^m$ | $AP^m_{50}$ | $AP^m_{75}$ |
|---|---|---|---|---|---|---|---|---|---|
| Mask R-CNN 1× schedule | PVT-M | 63.9M | − | 42.0 | 64.4 | 45.6 | 39.0 | 61.6 | 42.0 |
| | Swin-T | 47.8M | 264.0G | 42.2 | 64.6 | 46.2 | 39.1 | 61.6 | 42.0 |
| | Twins-PCPVT-S | 44.3M | 245.0G | 42.9 | 65.8 | 47.1 | 40.0 | 62.7 | 42.9 |
| | TransCNN-B | 46.4M | − | 44.0 | 66.4 | 48.5 | 40.2 | 63.3 | 43.2 |
| | ViL-M | 60.1M | 261.1G | 43.3 | 65.9 | 47.0 | 39.7 | 62.8 | 42.0 |
| | RegionViT-B | 92.2M | 287.9G | 43.5 | 66.7 | 47.4 | 40.1 | 63.4 | 43.0 |
| | RegionViT-B+ | 93.2M | 307.1G | 44.5 | 67.6 | 48.7 | 41.0 | 64.4 | 43.9 |
| | **CrossFormer-S** | 50.2M | 301.0G | **45.4** (+0.9) | 68.0 | 49.7 | **41.4** (+0.4) | 64.8 | 44.6 |
| | **CrossFormer-S‡** | 50.2M | 291.1G | **45.0** (+0.5) | 67.9 | 49.1 | **41.2** (+0.2) | 64.6 | 44.3 |
| | CAT-B | 71.0M | 356.0G | 41.8 | 65.4 | 45.2 | 38.7 | 62.3 | 41.4 |
| | PVT-L | 81.0M | 364.0G | 42.9 | 65.0 | 46.6 | 39.5 | 61.9 | 42.5 |
| | Twins-SVT-B | 76.3M | 340.0G | 45.1 | 67.0 | 49.4 | 41.1 | 64.1 | 44.4 |
| | ViL-B | 76.1M | 365.1G | 45.1 | 67.2 | 49.3 | 41.0 | 64.3 | 44.2 |
| | Twins-SVT-L | 119.7M | 474.0G | 45.2 | 67.5 | 49.4 | 41.2 | 64.5 | 44.5 |
| | Swin-S | 69.1M | 354.0G | 44.8 | 66.6 | 48.9 | 40.9 | 63.4 | 44.2 |
| | Swin-B | 107.2M | 496.0G | 45.5 | − | − | 41.3 | − | − |
| | **CrossFormer-B** | 71.5M | 407.9G | **47.2** (+1.7) | 69.9 | 51.8 | **42.7** (+1.4) | 66.6 | 46.2 |
| | **CrossFormer-B‡** | 71.5M | 398.1G | **47.1** (+1.6) | 69.9 | 52.0 | **42.7** (+1.4) | 66.5 | 46.1 |
| Mask R-CNN 3× schedule | PVT-M | 63.9M | − | 44.2 | 66.0 | 48.2 | 45.0 | 63.1 | 43.5 |
| | ViL-M | 60.1M | 261.1G | 44.6 | 66.3 | 48.5 | 40.7 | 63.8 | 43.7 |
| | Swin-T | 47.8M | 264.0G | 46.0 | 68.2 | 50.2 | 41.6 | 65.1 | 44.8 |
| | Shuffle-T | 48.0M | 268.0G | 46.8 | 68.9 | 51.5 | 42.3 | 66.0 | 45.6 |
| | **CrossFormer-S‡** | 50.2M | 291.1G | **48.7** (+1.9) | 70.7 | 53.7 | **43.9** (+1.6) | 67.9 | 47.3 |
| | PVT-L | 81.0M | 364.0G | 44.5 | 66.0 | 48.3 | 40.7 | 63.4 | 43.7 |
| | ViL-B | 76.1M | 365.1G | 45.7 | 67.2 | 49.9 | 41.3 | 64.4 | 44.5 |
| | Shuffle-S | 69.0M | 359.0G | 48.4 | 70.1 | 53.5 | 43.3 | 67.3 | 46.7 |
| | Swin-S | 69.1M | 354.0G | 48.5 | 70.2 | 53.5 | 43.3 | 67.3 | 46.6 |
| | **CrossFormer-B‡** | 71.5M | 398.1G | **49.8** (+1.3) | 71.6 | 54.9 | **44.5** (+1.2) | 68.8 | 47.9 |

Table 5: Semantic segmentation results on the ADE20K validation set. "MS IOU" means testing with variable input size.

| Semantic FPN (80K iterations) | | | | UPerNet (160K iterations) | | | | |
|---|---|---|---|---|---|---|---|---|
| Backbone | #Params | FLOPs | IOU | Backbone | #Params | FLOPs | IOU | MS IOU |
| PVT-M | 48.0M | 219.0G | 41.6 | Swin-T | 60.0M | 945.0G | 44.5 | 45.8 |
| Twins-SVT-B | 60.4M | 261.0G | 45.0 | Shuffle-T | 60.0M | 949.0G | 46.6 | 47.6 |
| Swin-S | 53.2M | 274.0G | 45.2 | **CrossFormer-S** | 62.3M | 979.5G | **47.6** (+1.0) | **48.4** |
| **CrossFormer-S** | 34.3M | 220.7G | **46.0** (+0.8) | **CrossFormer-S‡** | 62.3M | 968.5G | **47.4** (+0.8) | **48.2** |
| **CrossFormer-S‡** | 34.3M | 209.8G | **46.4** (+1.2) | Swin-S | 81.0M | 1038.0G | 47.6 | 49.5 |
| PVT-L | 65.1M | 283.0G | 42.1 | Shuffle-S | 81.0M | 1044.0G | 48.4 | 49.6 |
| CAT-B | 55.0M | 276.0G | 43.6 | **CrossFormer-B** | 83.6M | 1089.7G | **49.7** (+1.3) | **50.6** |
| **CrossFormer-B** | 55.6M | 331.0G | **47.7** (+4.1) | **CrossFormer-B‡** | 83.6M | 1078.8G | **49.2** (+0.8) | **50.1** |
| **CrossFormer-B‡** | 55.6M | 320.1G | **48.0** (+4.4) | Swin-B | 121.0M | 1088.0G | 48.1 | 49.7 |
| Twins-SVT-L | 103.7M | 397.0G | 45.8 | Shuffle-B | 121.0M | 1096.0G | 49.0 | − |
| **CrossFormer-L** | 95.4M | 497.0G | **48.7** (+2.9) | **CrossFormer-L** | 125.5M | 1257.8G | **50.4** (+1.4) | **51.4** |
| **CrossFormer-L‡** | 95.4M | 482.7G | **49.1** (+3.3) | **CrossFormer-L‡** | 125.5M | 1243.5G | **50.5** (+1.5) | **51.4** |

and 2K for validation. Similar to models for detection, we initialize the backbones with weights pre-trained on ImageNet, and MMSegmentation-based (Contributors, 2020) semantic FPN and UPer-Net (Xiao et al., 2018) are taken as the segmentation head. For FPN (Kirillov et al., 2019), we use an AdamW optimizer with learning rate and weight deacy of $1 \times 10^{-4}$. Models are trained for 80K iterations with batch size 16. For UPerNet, an AdamW optimizer with an initial learning rate of $6 \times 10^{-5}$ and a weight decay of 0.01 is used, and models are trained for 160K iterations.

**Results.** All results are shown in Table 5. Similar to object detection, CrossFormer exhibits a greater performance gain over the others when enlarging the model. For example, CrossFormer-T achieves 1.4% absolutely higher on IOU than Twins-SVT-B, but CrossFormer-B achieves 3.1% absolutely higher on IOU than Twins-SVT-L. Totally, CrossFormer shows a more significant advantage over the others on dense prediction tasks (*e.g.*, detection and segmentation) than on classification, implying that cross-scale interactions in the attention module are more important for dense prediction tasks than for classification.

Table 6: Results on the ImageNet validation set. The baseline model is CrossFormer-S (82.5%). We test with different kernel sizes of CELs.

| CEL's Kernel Size | | | | #Params/FLOPs | Acc. | $AP^m$ |
|---|---|---|---|---|---|---|
| *Stage-1* | *Stage-2* | *Stage-3* | *Stage-4* | | | |
| $4 \times 4$ | $2 \times 2$ | $2 \times 2$ | $2 \times 2$ | 28.3M / 4.5G | 81.5% | 39.7 |
| $8 \times 8$ | $2 \times 2$ | $2 \times 2$ | $2 \times 2$ | 28.3M / 4.5G | 81.9% | 40.2 |
| $4 \times 4, 8 \times 8$ | $2 \times 2, 4 \times 4$ | $2 \times 2, 4 \times 4$ | $2 \times 2, 4 \times 4$ | 30.6M / 4.8G | 82.3% | — |
| $4 \times 4, 8 \times 8, 16 \times 16, 32 \times 32$ | $2 \times 2, 4 \times 4$ | $2 \times 2, 4 \times 4$ | $2 \times 2, 4 \times 4$ | 30.7M / 4.9G | **82.5%** | **41.4** |
| $4 \times 4, 8 \times 8, 16 \times 16, 32 \times 32$ | $2 \times 2, 4 \times 4, 8 \times 8$ | $2 \times 2, 4 \times 4$ | $2 \times 2$ | 29.4M / 5.0G | 82.4% | — |

Table 7: Experimental results of ablation studies.

(a) Ablation studies on cross-scale embeddings (CEL) and long short distance attention (LSDA). The base model is CrossFormer-S (82.5%).

| PVT-like | Swin-like | LSDA | CEL | Acc. |
|---|---|---|---|---|
| ✓ | | | ✓ | **81.3%** |
| | ✓ | | ✓ | **81.9%** |
| | | ✓ | ✓ | **82.5%** |
| | | ✓ | | **81.5%** |

(b) Comparisons between different position representations. The base model is CrossFormer-S. Throughput is tested on $1\times$ V100 GPU.

| Method | #Params/FLOPs | Throughput | Acc. |
|---|---|---|---|
| APE | 30.9342M/4.9061G | 686 imgs/sec | 82.1% |
| RPB | 30.6159M/4.9062G | 684 imgs/sec | **82.5%** |
| DPB | 30.6573M/4.9098G | 672 imgs/sec | **82.5%** |
| DPB-residual | 30.6573M/4.9098G | 672 imgs/sec | 82.4% |

## 4.4 ABLATION STUDIES

**Cross-scale Embeddings vs. Single-scale Embeddings.** We conduct the experiments by replacing cross-scale embedding layers with single-scale ones. As we can see in Table 6, when using single-scale embeddings, the $8 \times 8$ kernel in *Stage-1* brings $0.4\%$ ($81.9\%$ vs. $81.5\%$) absolute improvement compared with the $4 \times 4$ kernel. It tells us that overlapping receptive fields help improve the model's performance. Besides, all models with cross-scale embeddings perform better than those with single-scale embeddings. In particular, our CrossFormer achieves $1\%$ ($82.5\%$ vs. $81.5\%$) absolute performance gain compared with using single-scale embeddings for all stages. For cross-scale embeddings, we also try several different combinations of kernel sizes, and they all show similar performance ($82.3\% \sim 82.5\%$). In summary, cross-scale embeddings can bring a large performance gain, yet the model is relatively robust to different choices of kernel size.

**LSDA vs. Other Self-attentions.** Two self-attention modules used in PVT and Swin are compared. Specifically, PVT sacrifices the small-scale features when computing the self-attention, while Swin restricts the self-attention in a local region, giving up the long-distance attention. As we can observe in Table 7a, compared against the PVT-like and Swin-like self-attention mechanisms, our Cross-Former outperforms them at least absolute $0.6\%$ accuracy ($82.5\%$ vs. $81.9\%$). The results show that performing the self-attention in a long-short distance manner is most conducive to improving the model's performance.

**DPB vs. Other Position Representations.** We compare the parameters, FLOPs, throughputs, and accuracies of the models among absolute position embedding (APE), relative position bias (RPB), and DPB. The results are shown in Table 7b. DPB-residual means DPB with residual connections. Both DPB and RPB outperform APE for absolute $0.4\%$ accuracy, which indicates that relative position representations are more beneficial than absolute ones. Further, DPB achieves the same accuracy ($82.5\%$) as RPB with an ignorable extra cost; however, as we described in Section 3.2.2, it is more flexible than RPB and applies to variable image size or group size. The results also show that residual connection in DPB does not help improve or even degrades the model's performance.

## 5 CONCLUSIONS

We proposed a novel transformer-based vision architecture, namely CrossFormer. Its core ingredients are Cross-scale Embedding Layer (CEL) and Long Short Distance Attention (LSDA), thereby yielding the cross-attention module. We further proposed a dynamic position bias, making the relative position bias apply to any input size. Extensive experiments show that CrossFormer achieves superior performance over other state-of-the-art vision transformers on several representative vision tasks. Particularly, CrossFormer is demonstrated to gain great improvements on object detection and segmentation, which indicates that CEL and LSDA are together essential for dense prediction tasks.

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

# A CROSSFORMER

## A.1 PSEUDO CODE OF LSDA

The pseudo code for LSDA is shown in Algorithm 1. As we can see, based on the vanilla self-attention module, both SDA and LDA are implemented with only ten lines of code, and only *reshape* and *permute* operations are used.

---
**Algorithm 1** LSDA code (PyTorch-like)

---
```
# H: height, W: width, G: group size of SDA/LDA
# x: input tensor (H, W, D)
class LSDA():
    def forward(x, type):
        ## group the embeddings
        if type == "SDA":
            x = x.reshaspe(H // G, G, W // G, G, D).permute(0, 2, 1, 3, 4)
        elif type == "LDA":
            x = x.reshaspe(G, H // G, G, W // G, D).permute(1, 3, 0, 2, 4)
        x = x.reshape(H * W // (G ** 2), G ** 2, D)

        ## the vanilla self-attention module
        x = Attention(x)

        ## un-group the embeddings
        x = x.reshaspe(H // G, W // G, G, G, D)
        if type == "SDA":
            x = x.permute(0, 2, 1, 3, 4).reshaspe(H, W, D)
        elif type == "LDA":
            x = x.permute(2, 0, 3, 1, 4).reshaspe(H, W, D)
        return x
```
---

## A.2 DYNAMIC POSITION BIAS (DPB)

Figure 4 gives an example of computing $(\Delta x_{ij}, \Delta y_{ij})$ with $G = 5$ in the DPB module. For a group of size $G \times G$, it is easy to deduce that:

$$0 \leq x, y < G$$
$$1 - G \leq \Delta x_{ij} \leq G - 1 \quad (3)$$
$$1 - G \leq \Delta y_{ij} \leq G - 1.$$

Thus, motivated by the relative position bias, we construct a matrix $\hat{B} \in \mathbb{R}^{(2G-1) \times (2G-1)}$, where

$$\hat{B}_{i,j} = DPB(1 - G + i, 1 - G + j), \ 0 \leq i, j < 2G - 1. \quad (4)$$

The complexity of computing $\hat{B}$ is $O(G^2)$. Then, the bias matrix $B$ in DPB can be drawn from $\hat{B}$, *i.e.*,

$$B_{i,j} = \hat{B}_{\Delta x_{ij}, \Delta y_{ij}}. \quad (5)$$

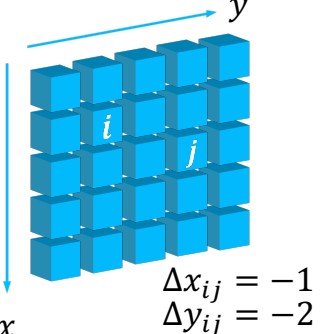

$$\Delta x_{ij} = -1$$
$$\Delta y_{ij} = -2$$

Figure 4: An example of computing $(\Delta x_{ij}, \Delta y_{ij})$.

When the image/group size (*i.e.*, $G$) is fixed, both $\hat{B}$ and $B$ will be also unchanged in the test phase. Therefore, we only need to compute $\hat{B}$ and $B$ once, and DPB is equivalent to relative position bias in this case.

## A.3 VARIANTS OF CROSSFORMER FOR DETECTION AND SEGMENTATION

We test two different backbones for dense prediction tasks. The variants of CrossFormer for dense prediction (object detection, instance segmentation, and semantic segmentation) are in Table 8. The architectures are the same as those for image classification except that different $G$ and $I$ in the first two stages are used. Notably, group size (*i.e.*, $G$ and $I$) does not affect the shape of weight tensors, so backbones pre-trained on ImageNet can be fine-tuned directly on other tasks even if they use different $G$ and $I$.

Table 8: CrossFormer-based backbones for object detection and semantic/instance segmentation. The example input size is $1280 \times 800$. $D$ and $H$ mean embedding dimension and the number of heads in the multi-head self-attention module, respectively. $G$ and $I$ are group size and interval for SDA and LDA, respectively.

| | Output Size | Layer Name | CrossFormer-T | CrossFormer-S | CrossFormer-B | CrossFormer-L |
|---|---|---|---|---|---|---|
| | | Cross Embed. | Kernel size: $4 \times 4, 8 \times 8, 16 \times 16, 32 \times 32$, Stride=4 | | | |
| Stage-1 | $320 \times 200$ | SDA/LDA MLP | $\begin{bmatrix} D_1=64 \\ H_1=2 \\ G_1=14 \\ I_1=16 \end{bmatrix} \times 1$ | $\begin{bmatrix} D_1=96 \\ H_1=3 \\ G_1=14 \\ I_1=16 \end{bmatrix} \times 2$ | $\begin{bmatrix} D_1=96 \\ H_1=3 \\ G_1=14 \\ I_1=16, \end{bmatrix} \times 2$ | $\begin{bmatrix} D_1=128 \\ H_1=4 \\ G_1=14 \\ I_1=16 \end{bmatrix} \times 2$ |
| | | Cross Embed. | Kernel size: $2 \times 2, 4 \times 4$, Stride=2 | | | |
| Stage-2 | $160 \times 100$ | SDA/LDA MLP | $\begin{bmatrix} D_2=128 \\ H_2=4 \\ G_2=14 \\ I_2=8 \end{bmatrix} \times 1$ | $\begin{bmatrix} D_2=192 \\ H_2=6 \\ G_2=14 \\ I_2=8 \end{bmatrix} \times 2$ | $\begin{bmatrix} D_2=192 \\ H_2=6 \\ G_2=14 \\ I_2=8 \end{bmatrix} \times 2$ | $\begin{bmatrix} D_2=256 \\ H_2=8 \\ G_2=14 \\ I_2=8 \end{bmatrix} \times 2$ |
| | | Cross Embed. | Kernel size: $2 \times 2, 4 \times 4$, Stride=2 | | | |
| Stage-3 | $80 \times 50$ | SDA/LDA MLP | $\begin{bmatrix} D_3=256 \\ H_3=8 \\ G_3=7 \\ I_3=2 \end{bmatrix} \times 8$ | $\begin{bmatrix} D_3=384 \\ H_3=12 \\ G_3=7 \\ I_3=2 \end{bmatrix} \times 6$ | $\begin{bmatrix} D_3=384 \\ H_3=12 \\ G_3=7 \\ I_3=2 \end{bmatrix} \times 18$ | $\begin{bmatrix} D_3=512 \\ H_3=16 \\ G_3=7 \\ I_3=2 \end{bmatrix} \times 18$ |
| | | Cross Embed. | Kernel size: $2 \times 2, 4 \times 4$, Stride=2 | | | |
| Stage-4 | $40 \times 25$ | SDA/LDA MLP | $\begin{bmatrix} D_4=512 \\ H_4=16 \\ G_4=7 \\ I_4=1 \end{bmatrix} \times 6$ | $\begin{bmatrix} D_4=768 \\ H_4=24 \\ G_4=7 \\ I_4=1 \end{bmatrix} \times 2$ | $\begin{bmatrix} D_4=768 \\ H_4=24 \\ G_4=7 \\ I_4=1 \end{bmatrix} \times 2$ | $\begin{bmatrix} D_4=1024 \\ H_4=32 \\ G_4=7 \\ I_4=1 \end{bmatrix} \times 2$ |

Table 9: Object detection results on COCO *val* 2017. "Memory" means the allocated memory per GPU reported by $torch.cuda.max\_memory\_allocated()$. ‡ indicates that models use different $(G, I)$ from classification models.

| Method | Backbone | $G_1$ | $I_1$ | $G_2$ | $I_2$ | Memory | #Params | FLOPs | $AP^b$ | $AP^b_{50}$ | $AP^b_{75}$ |
|---|---|---|---|---|---|---|---|---|---|---|---|
| RetinaNet $1\times$ schedule | CrossFormer-S | 7 | 8 | 7 | 4 | 14.7G | 40.8M | 282.0G | 44.4 | 65.8 | 47.4 |
| | CrossFormer-S‡ | 14 | 16 | 14 | 8 | 11.9G | 40.8M | 272.1G | 44.2 | 65.7 | 47.2 |
| | CrossFormer-B | 7 | 8 | 7 | 4 | 22.8G | 62.1M | 389.0G | 46.2 | 67.8 | 49.5 |
| | CrossFormer-B‡ | 14 | 16 | 14 | 8 | 20.2G | 62.1M | 379.0G | 46.1 | 67.7 | 49.0 |
| Mask-RCNN $1\times$ schedule | CrossFormer-S | 7 | 8 | 7 | 4 | 15.5G | 50.2M | 301.0G | 45.4 | 68.0 | 49.7 |
| | CrossFormer-S‡ | 14 | 16 | 14 | 8 | 12.7G | 50.2M | 291.1G | 45.0 | 67.9 | 49.1 |
| | CrossFormer-B | 7 | 8 | 7 | 4 | 23.8G | 71.5M | 407.9G | 47.2 | 69.9 | 51.8 |
| | CrossFormer-B‡ | 14 | 16 | 14 | 8 | 21.0G | 71.5M | 398.1G | 47.1 | 69.9 | 52.0 |

Table 10: Semantic segmentation results on ADE20K validation set with semantic FPN or UPerNet as heads.

| Backbone | $G_1$ | $I_1$ | $G_2$ | $I_2$ | Semantic FPN (80K iterations) | | | | UPerNet (160K iterations) | | | | |
|---|---|---|---|---|---|---|---|---|---|---|---|---|---|
| | | | | | Memory | #Params | FLOPs | IOU | Memory | #Params | FLOP | IOU | MS IOU |
| CrossFormer-S | 7 | 8 | 7 | 4 | 20.9G | 34.3M | 220.7G | 46.0 | – | 62.3M | 979.5G | 47.6 | 48.4 |
| CrossFormer-S‡ | 14 | 16 | 14 | 8 | 20.9G | 34.3M | 209.8G | 46.4 | 14.6G | 62.3M | 968.5G | 47.4 | 48.2 |
| CrossFormer-B | 7 | 8 | 7 | 4 | 14.6G | 55.6M | 331.0G | 47.7 | 15.8G | 83.6M | 1089.7G | 49.7 | 50.6 |
| CrossFormer-B‡ | 14 | 16 | 14 | 8 | 14.6G | 55.6M | 320.1G | 48.0 | 15.8G | 83.6M | 1078.8G | 49.2 | 50.1 |
| CrossFormer-L | 7 | 8 | 7 | 4 | 25.3G | 95.4M | 497.0G | 48.7 | 18.1G | 125.5M | 1257.8G | 50.4 | 51.4 |
| CrossFormer-L‡ | 14 | 16 | 14 | 8 | 25.3G | 95.4M | 482.7G | 49.1 | 18.1G | 125.5M | 1243.5G | 50.5 | 51.4 |

# B EXPERIMENTS

## B.1 OBJECT DETECTION

Table 9 provides more results on object detection with RetinaNet and Mask-RCNN as detection heads. As we can see, a smaller $(G, I)$ achieves a higher AP than a larger one, but the performance gain is marginal. Considering that a larger $(G, I)$ can save more memory cost, we think $(G_1 = 14, I_1 = 16, G_2 = 14, I_2 = 8)$, which accords with configurations in Table 8, achieves a better trade-off between the performance and cost.

Table 11: Classification results on ImageNet dataset after plugging CEL into other vision transformers.

| Models | #Params | FLOPs | Accuracy |
|---|---|---|---|
| Swin | 29.0M | 4.5G | 81.3% |
| Swin + CEL | 29.2M | 4.8G | **81.9%** (+0.6%) |
| NesT | 38.4M | 10.4G | 83.3% |
| NesT + CEL | 38.5M | 10.6G | **83.8%** (+0.5%) |

### B.2 SEMANTIC SEGMENTATION

Similar to object detection, we test two different configurations of $(G, I)$ for semantic segmentation's backbones. The results are shown in Table 10. As we can see, the memory costs of the two configurations are almost the same, which is different from experiments on object detection. Further, when taking semantic FPN as the detection head, CrossFormers‡ show advantages over CrossFormers on both IOU (*e.g.*, $46.4$ vs. $46.0$) and FLOPs (*e.g.*, $209.8G$ vs. $220.7G$). When taking UPerNet as the segmentation head, a smaller $(G, I)$ achieves higher performance like object detection.

### B.3 CLASSIFICATION WITH CEL

Table 11 shows results on ImageNet dataset after plugging CEL into other vision transformers. As we can see, CEL brings about $0.5\%$ performance gain for both Swin Transformer and NesT, which further shows the effectiveness of CEL.

## C DISCUSSION

### C.1 CROSS-SCALE FEATURE EXTRACTION

**CEL vs. GoogLeNet/MixConv.** Since GoogLeNet (Szegedy et al., 2015) and MixConv (Tan & Le, 2019) use multi-scale convolutional kernels at every layer of models, the largest kernel size can only be $7 \times 7$, and larger kernel size will induce unaccpetable computational budget. In contrast, CEL only introduces cross-scale features at the start of each stage (at most 4 layers), so kernels with very large size (e.g., $32 \times 32$) can also be used for diversified-size objects.

**CEL vs. CoaT/CViT.** Some vision transformers also take advantage of cross-scale features. For example, CoaT (Xu et al., 2021a) only uses multi-scale features at later layers by mixing features from front layers, while CEL introduces multi-scale features at the start of each stage, thus tokens embeddings at all layers (not only those at later layers) could be seen as cross-scale. Besides, CViT (Chen et al., 2021a) keeps embeddings of different scales in different branches and introduces cross-scale interaction through self-attention between branches. This method can well preserve the information of each scale, but multiple branches take more memory/computational/parameters cost than a single-branch model. As a result, for a controllable cost, CViT can only introduce two different scales ($4 \times 4$ and $8 \times 8$). In contrast, CEL only appears at the start of each stage (at most 4 layers). So, CEL can introduce more different scales (e.g., $4 \times 4, 8 \times 8, 16 \times 16, 32 \times 32$) than CViT with ignorable extra cost ($< 0.1M$ parameters and $< 0.3G$ FLOPs in most cases).

### C.2 SPARSE SELF-ATTENTION AND GROUP CONVOLUTION

**LSDA vs. ShuffleNet/IGCNet.** LSDA, ShuffleNet (Zhang et al., 2018b) and IGCNets (Zhang et al., 2017) devote to reducing the computational cost by approximating the original operations (self-attention and convolution) in a lower cost. Nevertheless, they are essentially different: (1) **Different target and dimension**: ShuffleNet/IGCNet devotes to reducing the budget caused by the large channels of convolution. So, both ShuffleNet and IGCNet do shuffle-and-permutation along channel dimension. While for the self-attention module, the budget mainly attributes to the large spatial size of feature maps (i.e., the large number of embeddings). Thus, LSDA is mainly operating along the spatial dimension. (2) **Different motivation**: During the process of designing LSDA, we first consider the locality of images and propose to extract short-distance attention through SDA.

Meanwhile, to preserve the ability of modeling long-distance dependency, LDA is further proposed to fill this gap. In contrast, ShuffleNet/ICGNets do not need to consider these aspects because all channels are equivalent, and locality does not exist along channel dimmension. Though they also perform convolution among adjacent channels first, it is for the sake of fast memory access.

**LSDA vs. Other Sparse Self-attention.** Previous works proposed some sparse self-attention, such as Big Bird (Zaheer et al., 2020), Longformer (Beltagy et al., 2020), Sparse Transformer (Child et al., 2019), etc. Wherein, as a generative model, Sparse Transformer proposed a strided attention in the one-dimensional case, and every pixel can see the pixel generated ahead of itself. Instead, LSDA works in two-dimensional case. Further, LSDA has a co-design with CEL. As we described in Section 3.2.1 and Figure 3, in long-distance attention (LDA), the small-scale patches of two embeddings are non-adjacent, so it is difficult to judge their relationship without the help of the context. In other words, it will be hard to build the dependency between these two embeddings if they are only constructed by small-scale patches (i.e., single-scale feature). On the contrary, adjacent large-scale patches provide sufficient context to link these two embeddings, which makes long-distance cross-scale attention easier and more meaningful.

