# OpenReview forum: "CrossFormer: A Versatile Vision Transformer Hinging on Cross-scale Attention"
_ICLR.cc/2022/Conference — ICLR 2022 Poster_

### Official Review · Reviewer_BmmR · 2021-10-29

**Correctness:** 3
**Technical Novelty And Significance:** 3
**Empirical Novelty And Significance:** 3
**Recommendation:** 8
**Confidence:** 4

**Main Review:**

## Strengths
* Overall simplicity and elegance of the introduced components. I think the paper strikes a good balance between introduced complexity and achieved performance.
* Results are very strong on a number of vision tasks, and in particular on dense prediction tasks
* Well presented and easy to understand.

## Weaknesses
* Context to prior work:
  * In principle a fully connected layer over a large patch can learn to represent patches of multiple smaller scales as well. The Cross-scale embedding layer in this paper is essentially just a more constraint version of it. Taking 32x32 patches (4x4 stride) and projecting those to D is strictly more expressive than the procedure described in the paper. This is not to say that it is not useful as it clearly provides inductive bias, but it is less expressive which is the opposite of what the paper says. Hence I don't agree with the blank statement: "They fail to build the interactions among features of different scales, whereas such an ability is very vital for a lot of vision tasks." A more subtle discussion on inductive biases would be more appropriate and I would ask the authors to change that. Personal side-note: A more interesting take on scale (IMO) would be to introduce a mechanism that enforces scale invariance (at least in some range).
  * Long-short distance attention is basically a version of sparse attention (strided attention) that was introduced for instance in [1]. There are also other papers, e.g., [2, 3], that introduced very similar sparse attention mechanisms, however, mostly with a focus on text but they easily generalize to images. So the novelty of the presented mechanism here is limited (at best) and the absence of proper references to similar work is a bit concerning. A more thorough study and comparison with prior work should be added.
* Novelty (Framing): The individual components are not very novel (see points before), and it should be made clear in the paper. It's, however, not a big issue for me because finding the right combination of these components in order to strike the right balance between complexity and performance is more important and as such also novel.

## Questions
* Ablations on a dense prediction task to see where the improvements really come from. The authors hypothesize they are due to cross scale patches. These ablations are important to also verify that it isn't the overall training setup or other subtle differences in the architecture that give the advantage.
* A strange choice is that smaller patches project to larger dimensions, leading to larger compression. Was this a deliberate choice and ablated?
* Setting G and I to different values and not making G divisible by I is a bit odd, I wonder why such values were chosen.

* [1] https://arxiv.org/abs/1904.10509
* [2] https://arxiv.org/abs/2007.14062
* [3] https://arxiv.org/abs/2004.05150
* [4] https://arxiv.org/abs/2103.15358

**Summary Of The Paper:**

The paper describes a pyramidal vision transformer that introduces cross-scale patch embeddings, dynamic relative position biases and "Long-Short Attention". The model shows superior performance on imagenet classification and especially on dense prediction tasks. Ablations on classification are provided hinting that the multi-scale embeddings are the most important factor contributing to the overall superior results to similar, previous architectures.

**Summary Of The Review:**

The paper is overall well written and easy to understand. I think there are some strong contributions that are a bit oversold (in my view). However, I am leaning towards accepting the paper given that some of the concerns and questions are properly addressed. I am willing to raise my score in case those concerns are addressed.

UPDATE after rebuttal. I am willing to raise my score after reading the rebuttal and the updates to the paper. I think the paper is a good and clean contribution as it is conceptually simpler (in my view) than most of the previous works while showing stronger performance.

---

> ### Author Response · Authors · 2021-11-19
> **Responses to "the context to prior work"**
>
> ### Glossary:
>
> - CEL: Cross-scale Embedding Layer proposed in our paper, which appears at the start of each stage and provides cross-scale embeddings for the model.
>
> - LSDA: Long Short Distance Attention proposed in our paper, which applies SDA and LDA alternately for short distance and long distance attentions, respectively.
>
> ### Responses:
>
> 1. **The claim about CEL**: We agree with your point view that cross-scale embedding is more like an inductive bias, which enforces some dimensions to focus on small scale features only. Specifically, the previous methods of generating token embeddings often fall into two schemes:
>
>    1. **Small non-overlapping patches**: An image of size $H \times W$ is sampled with a kernel of size $P \times P$ and stride of $P\times P$ (e.g., $P=4$), yielding $\frac{H}{P}\times\frac{W}{P}$ patches.
>    2. **Large overlapping patches**: An image of size $H \times W$ is sampled with a kernel of size $P' \times P'$ and stride $P \times P$ (e.g., $P=4, P'=8$), yielding $\frac{H}{P}\times\frac{W}{P}$ patches. In this case, each patch has $P'-P$ overlap with its adjacent patch.
>
>    Theoretically, patches for the second scheme can capture features of any scale that $\le P'$, which owns higher capacity than the first scheme. In the extreme case, it can only focus on the $P \times P$ region for small-scale features, which is the same as the first scheme. Experiments in Table 6 of the paper also show that the second scheme performs better than the first one.
>
>    We update our claim in the introduction as follows: existing vision transformers do not explicitly utilize features of different scales, whereas multi-scale features are very vital for a lot of vision tasks. The reasons are: the embeddings are generated from equal-sized patches. Though these patches theoretically have a chance to extract any scale features if only the receptive field is large enough, it is difficult to promise that they can learn appropriate multi-scale features automatically in practice. CEL enforces some dimensions (e.g., dimensions from $4 \times 4$ patches) to focus on small-scale features only, while others (e.g., those from $8 \times 8$ patches) have a chance to learn large-scale features, leading to an embedding with explicitly cross-scale features.
>
> 2. **LSDA vs. other sparse attentions**: Recently, many different self-attention mechanisms are proposed in recent vision transformers, such as Swin, Shuffle Transformer, CvT, etc. Most of them can be seen as a special case of sparse attention, including LSDA in CrossFormer. While the motivations and specific attention mechanisms behind them may be different:
>
>    1. **LSDA:** Considering the locality of images, it first extract short-distance attention through SDA. Meanwhile, to preserve the ability of modeling long-distance dependency, LDA is further proposed to fill this gap. Furthermore, LSDA has a co-design with CEL. As we described in Section 3.2.1 and Figure 3, in long-distance attention (LDA), the small-scale patches of two embeddings are non-adjacent, so it is difficult to judge their relationship without the help of the context. In other words, it will be hard to build the dependency between these two embeddings if they are only constructed by small-scale patches (i.e., single-scale feature). On the contrary, adjacent large-scale patches provide sufficient context to link these two embeddings, which makes long-distance cross-scale attention easier and more meaningful.
>    2. **Longformer and Sparse Transformer:** Longformer [2] and Sparse Transformer [1] propose strided attention in the one-dimensional case. Instead, LSDA works in a two-dimensional case, which allows us to fully utilize the locality of images.
>    2. **BigBird:** Bigbird [3] uses a combination of local window/glocal/random attention, which is different from LSDA.
>
>    Due to limited space, we have to sketch these discussions in Section 2, and detail it in the appendix C.1.
>
> [1] Generating Long Sequences with Sparse Transformers.
>
> [2] Longformer: The Long-Document Transformer.
>
> [3] Big Bird: Transformers for Longer Sequences.

---

> > ### Comment · Reviewer_BmmR · 2021-11-22
> > **RE**
> >
> > I guess the distinction between 1D and 2D here is just a detail and I wouldn't claim novelty over extending some sparse attention mechanisms to 2D. However, as mentioned I don't have a real problem with that.
> >
> > Thanks for acknowledging that the CEL is just a special case of path embeddings that seems to help learning.

---

> ### Author Response · Authors · 2021-11-19
> **Responses to the questions**
>
> ### Glossary:
>
> - CEL: Cross-scale Embedding Layer proposed in our paper, which appears at the start of each stage and provides cross-scale embeddings for the model.
>
> ### Responses
> 1. **Ablations on dense prediction tasks**: To show the importance of CEL to dense prediction tasks, we replace CEL in CrossFormer with single-scale embedding layer (just like models shown in Table 6 of the paper), pretrain the model on ImageNet, and fine-tune the model on dense prediction tasks (object detection and instance segmentation). Results are in the following table. As we can see, models with CEL perform best (about 1% higher than the others in AP$^m$ or AP$^b$).
>
>    | Stage-1's CEL kernel                                      | Stage-2's CEL kernel       | Stage-3's CEL kernel       | Stage-4's CEL kernel       | RetinaNet AP$^b$ | Mask R-CNN AP$^b$ | Mask R-CNN AP$^m$ |
>    | --------------------------------------------------------- | -------------------------- | -------------------------- | -------------------------- | ---------------- | ----------------- | ----------------- |
>    | $4\times4$                                                | $2 \times 2$               | $2 \times 2$               | $2 \times 2$               | 42.0             | 42.9              | 39.7              |
>    | $8 \times 8$                                              | $2 \times 2$               | $2 \times 2$               | $2 \times 2$               | 42.8             | 43.5              | 40.2              |
>    | $8 \times 8$                                              | $4 \times 4$               | $4 \times 4$               | $4 \times 4$               | 43.5             | 44.3              | 40.6              |
>    | $4\times4$, $8 \times 8$, $16 \times 16$, $32 \times 32$ | $2 \times 2$, $4 \times 4$ | $2 \times 2$, $4 \times 4$ | $2 \times 2$, $4 \times 4$ | **44.4**         | **45.4**          | **41.4**          |
>
> 2. **Why are smaller patches projected to larger dimensions**: We thought of three projecting rules:
>
>    1. Projecting smaller patches to larger dimensions while larger patches to smaller dimensions (i.e., the one in the paper).
>    2. Partitioning the dimension evenly.
>    2. Projecting larger patches to larger dimensions and smaller patches to smaller dimensions
>
>    Finally, we did not choose the second/third schemes for two reasons:
>
>    1. The core of CEL is mixing features of different scales, which should support any number of scales for generality. However, the dimension cannot be partitioned evenly (the second scheme) if the total dimension is not divisible by the number of different scales. For example, supposing that the total dimension is 128, and we only want three different scales of patches (4x4, 8x8, and 16x16), the dimension cannot be partitioned evenly.
>    2. The second/third schemes takes more FLOPs than the first one, while they achieve similar accuracy. The following table takes CrossFormer-S and a $224 \times 224$ input as an example (A larger total dimension or input image will further lead to a larger FLOPs difference).
>
>    | Scheme   | Totally Dims | 4x4 Patchs | 8x8 Patchs | 16x16 Patchs | 32x32 Patchs | FLOPs    | Accuracy |
>    | -------- | ------------ | ---------- | ---------- | ------------ | ------------ | -------- | -------- |
>    | Scheme-1 | 96           | 48 dims    | 24 dims    | 12 dims      | 12 dims      | **4.9G** | **82.5** |
>    | Scheme-2 | 96           | 24 dims    | 24 dims    | 24 dims      | 24 dims      | 5.1G     | **82.5** |
>    | Scheme-3 | 96           | 12 dims    | 12 dims    | 24 dims      | 48 dims      | 5.3G     | 82.4     |
>
> 3. **The choice of $G$ and $I$:**
>
>    1. $G$ is not necessarily divisible by $I$, but both $G$ and $I$ should be divisible by the input/output size (i.e., $S$ in Table 1 of the paper) of each layer because $S=G \times I$, or the input must be padded.
>    2. The common input size for classification on ImageNet is $224 \times 224$. So, as shown in Table 1 of the paper, the input sizes for four stages are $56 \times 56, 28\times28, 14\times14,$ and $7\times7$, respectively. That is, $S=[56, 28, 14, 7]$ for four stages. Choosing $G$ as $7$ makes $S$ always divisible by $G$, and no padding is needed.

---

### Official Review · Reviewer_K7wS · 2021-11-01

**Correctness:** 3
**Technical Novelty And Significance:** 2
**Empirical Novelty And Significance:** 2
**Recommendation:** 5
**Confidence:** 5

**Main Review:**

Pros:
- How to build the interaction among different scale features is an interesting problem in computer vision. The proposed method provides an easy way to build the multi-scale feature interaction for transformer-based architecture.
- Comprehensive experiments are conducted, and the results showed the effectiveness of the proposed architecture.
- The idea of the paper is clear and easy to follow.

Cons:
- The short-distance attention is essentially the local window attention in the Swin-transformer, which is not new.
- The results in Table 6(b) showed that the proposed dynamic positional embedding has a similar performance with relative positional embedding. It's hard to say that the proposed dynamic positional embedding (DPE) is a contribution. The original relative positional embedding is also learned and dynamic and is also the function of relative coordinates. So, the problem becomes how to define the function.  However, the authors did not show how and why the function in the proposed DPE is superior to original relative positional embedding.
- The way to generate the multi-scale features is not novel. In the previous works [3, 4], the mixed convolution, which is especially the same as the proposed Cross-scale Embedding Layer.
- The long-distance attention is like the method of interleaved group or shuffle, which are proposed in the paper of [1,2] over the spatial features. I think the authors should carefully discuss the relation between them.

[1] ShuffleNet: An Extremely Efficient Convolutional Neural Network for Mobile Devices

[2] Interleaved Group Convolutions

[3] Going Deeper with Convolutions

[4] MixConv: Mixed Depthwise Convolutional Kernels


**Summary Of The Paper:**

The paper proposed a novel multi-stage vision transformer architecture, named CrossFormer, which aimed at building interaction among features from different scales. The proposed architecture follows the layout of Swin transformer but replaces the shifted windows transformer block with the proposed short-distance attention and long-distance attention. A cross-scale module is proposed to replace the patch embedding or patch merging layer in the Swin transformer. Comprehensive experiments are conducted on image classification, objection detection, and segmentation tasks. Results shows the effectiveness of the proposed architecture.

**Summary Of The Review:**

The idea of the paper is clear. Comprehensive experiments are performed, and superior results over several benchmarks demonstrated the effectiveness of the proposed architecture. My main concern is that the idea of the proposed components is not new in the CNN-based works. The overall novelty of the paper is limited.

---

> ### Author Response · Authors · 2021-11-19
> **Responses to the concerns**
>
> ### Glossary:
> - CEL: Cross-scale Embedding Layer proposed in our paper, which appears at the start of each stage and provides cross-scale embeddings for the model.
> - LSDA: Long Short Distance Attention proposed in our paper, which applies SDA and LDA alternately for short distance and long distance attentions, respectively.
> - DPB: Dynamic Position Bias, a new position representation which not only supports dynamic image/group size, but also achieves similar accuracy to the popular relative position bias (RPB).
>
> ### Responses:
> 1. **Short distance attention vs. local window attention**: SDA and LDA work as a whole, which comprise our self-attention mechanism (i.e., LSDA). Though SDA is equivalent to the local window attention in Swin Transformer, our motivation and used self-attention mechanism are different from Swin. Specifically, Swin's self-attention consists of a local window and a shifted local window, both of which are for local attentions. While we think both short-distance/local and long-distance dependencies are important, so short distance attention (SDA) and long distance attention (LDA) appear alternately in the model.
> 2. **DPB vs. RPB**:
>    1. **Advantages of DPB over RPB**: The comparison between DPB and RPB is shown in the following table. The main contribution about DPB is **supporting dynamic group size**. Please kindly refer to our responses to #R1 (DPB vs. RPB) for more details.
> | Methods    | Fixed Group Size (E.g., Swin Transformer)                    | Dynamic Group Size (E.g., PVT, CrossFormer, RegionViT, etc.) |
> | ---------- | ------------------------------------------------------------ | ------------------------------------------------------------ |
> | RPB        | Applicable                                                   | **NOT** applicable                                           |
> | DPB (ours) | Applicable, equivalent to RPB in the test phase (shown in the appendix A.2) | Applicable                                                   |
> 3. **Comparison between CrossFormer and cross-scale CNNs**: Many previous CNNs also take advantage of features of different scales. However, our CEL mainly has two differences from them:
>
>    1. **CEL supports very large kernel size**: Since [1,2] use multi-scale convolutional kernels at every layer of models, the largest kernel size can only be 7x7, and larger kernel size will induce an unacceptable computational budget. In contrast, we only introduce cross-scale features at the start of each stage (at most 4 layers), so kernels with very large size (e.g., 32x32) can also be used for diversified-size objects.
>    2. **CEL has a co-design with LDA**: As we described in Section 3.2.1 and Figure 3, in long-distance attention (LDA), the small-scale patches of two embeddings are non-adjacent, so it is difficult to judge their relationship without the help of the context. In other words, it will be hard to build the dependency between these two embeddings if they are only constructed by small-scale patches (i.e., single-scale feature). On the contrary, adjacent large-scale patches provide sufficient context to link these two embeddings, which makes long-distance cross-scale attention easier and more meaningful.
>
>    Due to limited space, we have to sketch these discussions in Section 2, and detail it in the appendix C.1.
>
> 4. **Comparsion between LSDA and ShuffleNet / IGCNets**: Both LSDA and ShuffleNet[3] / IGCNets[4] devote to reducing the computational cost by approximating the original operations (self-attention and convolution) in a lower cost. Nevertheless, they are essentially different:
>
>    1. **Different target and dimension:** ShuffleNet/IGCNet devotes to reducing the budget caused by the large channels of convolution. So, both ShuffleNet and IGC do shuffle-and-permutation along channel dimension. While for the self-attention module, the budget mainly attributes to the large spatial size of feature maps (i.e., the large number of embeddings). Thus, LSDA is mainly operating along the spatial dimension.
>    2. **Different motivation:** During the process of designing LSDA, we first consider the locality of images and propose to extract short-distance attention through SDA. Meanwhile, to preserve the ability of modeling long-distance dependency, LDA is further proposed to fill this gap. In contrast, ShuffleNet/ICGNets do not need to consider these aspects because all channels are equivalent, and locality does not exist along channel dimension. Though they also perform convolution among adjacent channels first, it is for the sake of fast memory access.
>
>    Due to limited space, we have to sketch these discussions in Section 2, and detail it in the appendix C.1.
>
> [1] Going Deeper with Convolutions. CVPR 2015
>
> [2] MixConv: Mixed Depthwise Convolutional Kernels. BMVC 2019
>
> [3] ShuffleNet: An Extremely Efficient Convolutional Neural Network for Mobile Devices. CVPR 2018
>
> [4] Interleaved Group Convolutions for Deep Neural Networks. ICCV 2017

---

> ### Comment · Reviewer_K7wS · 2021-11-30
> **Final recommendation**
>
> The authors solved my concerns. I'd like to upgrade to weak acceptance.

---

> > ### Author Response · Authors · 2021-11-30
> > **Thanks for your reply**
> >
> > Thanks for your reply, and we are glad that our responses solve your concerns. We kindly remind you that you may need to **update the score accordingly**.

---

### Official Review · Reviewer_euuQ · 2021-11-07

**Correctness:** 3
**Technical Novelty And Significance:** 3
**Empirical Novelty And Significance:** Not applicable
**Recommendation:** 6
**Confidence:** 4

**Main Review:**

Pros:
1. The idea of CEL is sound and its design is neat. As this layer can be plugged in other structures easily, it can also boost other models.
2. Thorough experiments verify the effectiveness of the proposed techniques. And CrossFormer achieve good performance on several vision taks.

Cons:
1. Some comparisons and discussions with related works are missed. For example, CrossViT[1] also combine features from patches of different sizes and Shuffle Transformer[2] proposes a spatial shuffle mechanism for transformer layer, which is similar with the long attention module in this paper.

[1]  Chen, C. F., Fan, Q., & Panda, R. (2021). Crossvit: Cross-attention multi-scale vision transformer for image classification. arXiv preprint arXiv:2103.14899.

[2]  Huang, Z., Ben, Y., Luo, G., Cheng, P., Yu, G., & Fu, B. (2021). Shuffle Transformer: Rethinking Spatial Shuffle for Vision Transformer. arXiv preprint arXiv:2106.03650


**Summary Of The Paper:**

This paper proposes two new techniques: Cross-scale Embedding Layer (CEL) and Long Short Distance Attention (LSDA). Cross-scale Embedding Layer samples patches with convolution kernels of different sizes and concatenates them as one embedding, so that patch embeddings contain information of multiple scales. Long Short Distance Attention includes a short attention module and a long attention module. The short attention is computed in a local region. The long one covers the whole image but is computed sparsely. These two new designs enable the network to learn cross-scale features. Also, a dynamic position bias is designed to make the network suitable for inputs of different resolutions.

**Summary Of The Review:**

Overall, the ideas of CEL and LSDA are sound and solid experiments support most claims in the paper. However, some designs are not new and some related works are missed. I would rate this as a borderline paper (marginally above the acceptance threshold).

---

> ### Author Response · Authors · 2021-11-19
> **Comparisons and discussions with related works**
>
> ### Glossary:
>
> - CEL: Our proposed Cross-scale Embedding Layer, which appears at the start of each stage and provides cross-scale embeddings for the model.
>
> - LSDA: Our proposed Long Short Distance Attention, which applies SDA and LDA alternately for short distance and long distance attentions, respectively.
>
> ### Responses:
>
> The experimentally comparison with CrossViT and Shuffle Transformer can be seen in Table 1, 4, and 5 of our paper (We term CrossViT as CViT in our paper). The theoretical discussion about CrossViT and ShuffleTransformer is supplemented in the appendix as below:
>
> 1. **Comparison between CEL and CrossViT's cross-scale interaction:** CrossViT also introduces interactions among features of different scales, but goes in a different way from CrossFormer:
>    1. CrossViT keeps embeddings of different scales in different branches, and introduces cross-scale interaction through self-attention between branches. This method can well preserve the information of each scale, but multiple branches take greater memory/computational/parameters cost than a single-branch model. As a result, for keeping a controllable cost, CrossViT can only introduce two different scales (4x4 and 8x8).
>    2. CrossFormer introduces cross-scale features through CEL layers. CEL directly mixes features of different scales at the start of each stage. It only induces an ignorable extra cost (<0.1M parameters and <0.3G FLOPs in most cases) at CEL layers. So, we can introduce more different scales (e.g., 4x4, 8x8, 16x16, 32x32) than CrossViT.
> 2. **Comparison between LSDA and Shuffle Transformer's attention:** Both LSDA and Shuffle Transformer propose an approximation of the vanilla self-attention. Shuffle Transformer simply takes all embeddings as a one-dimensional sequence. Embeddings are grouped randomly (i.e., shuffle-and-permutation). As opposed, LSDA considers embeddings in a two-dimensional case, which allows us to fully utilize the properties (e.g., locality) of images. Specifically, considering the locality of images, it first extracts short-distance attention through SDA. Meanwhile, to preserve the ability of modeling long-distance dependency, LDA is further proposed to fill this gap. The experiments on classification, detection, and segmentation also show our advantages over Shuffle Transformer.
>
> Due to limited space, we have to sketch this discussion in Section 2, and detail it in the appendix C.1.
>
> [1] CrossViT: Cross-Attention Multi-Scale Vision Transformer for Image Classification. ICCV 2021
>
> [2] Shuffle Transformer: Rethinking Spatial Shuffle for Vision Transformer. Arxiv 2021

---

> > ### Comment · Reviewer_euuQ · 2021-11-30
> > **RE**
> >
> > Thanks for authors' response. The response has solved some of my concerns. I would keep my rating score as weak accept.

---

### Official Review · Reviewer_GtDH · 2021-11-08

**Correctness:** 3
**Technical Novelty And Significance:** 3
**Empirical Novelty And Significance:** 3
**Recommendation:** 6
**Confidence:** 5

**Main Review:**

Pros:
1. The Cross-scale Embedding Layer (CEL) extends the single-scale feature maps used in prior works of ViTs (e.g. ViT, DeiT, PVT, Swin)  to multi-scale feature maps. The mixture of multi-scale features enable a richer context, especially for dense prediction tasks like object detection, instance segmentation and semantic segmentation. In addition, the CEL employs a series of kernel with the same stride and decreasing number of output dimensions, which is able to control the total computation budgets.

2. In the self-attention mechanism, the proposed CrossFormer introduces a long short distance attention (LSDA), which includes a long distance attention (LDA) and a short distance attention (SDA). The SDA is very similar to the shifted window attention in Swin, which splits feature maps into windows (denoted by groups in this paper) and performs self-attention in each window (group). Different from Swin, the LDA module has a dilated window where each embedding in the window has a large interval to its adjacent embeddings. The proposed LDA can model long range interactions while still keeping a linear complexity in computation.

3. In the experiment section, the proposed approach achieves state-of-the-art performance on all tasks. Specifically, on dense prediction task like detection and segmentation, the proposed CrossFormer obtains decent improvement over prior works with similar computation budget.

Cons / Questions:
1. The proposed CrossFormer focuses on better modeling of cross-scale information for ViTs: CEL enables multi-scale features and LSDA enables long distance interaction (which implies interaction among large-scale features). However, there are already some works on the cross-scale information such as [1]. Although the prior works may not use the identical way, it is still needed to have some discussion in the manuscript to compare the alternatives to model cross-scale information and the benefits from the proposed CrossFormer.

    [1] Co-Scale Conv-Attentional Image Transformers, Xu et al, ICCV 2021

2. The proposed LSDA may have some limitations in its design: Table 1 shows that each stage in CrossFormer has the same choice of (G, I) and Appendix A.1 shows that G*I is the size of the feature map. Thus, when the size is large (e.g. stage 1, size=56), since G is alway 7, interval I will be large (e.g. stage 1, I=7), which causes the elements in the window (group) are very scattered, and the behavior of the LDA could be very different from SDA in the same stage. I wonder if such large gap between LDA and SDA could lead to some potential issues. In addition, perhaps ablation study on LDA with lower interval is needed, since it may work better together with SDA.

3. This paper also proposes a dynamic position bias (DPB), which aims to replace the relative position bias (RPB) used in Swin and early works in NLP (Shaw et al). However, Table 6(b) does not show that the proposed DPB is able to outperform RPB, which limits the use of DPB. In addition, it is claimed that DPB has the capability to support different window (group) size. It would be better to have a case that demonstrates this benefit.

4. In the empirical experiments, on the image classification task, the proposed CrossFormer obtains similar performance as the best of prior works. Thus, I wonder if it is possible to apply the proposed CEL and LSDA to prior works (e.g. RegionViT or NesT) to further improve the performance?


**Summary Of The Paper:**

This paper proposes a novel vision transformer architecture called CrossFormer which focuses on the cross-scale ability in the attention module. Specifically, the proposed CrossFormer introduce Cross-scale Embedding Layer (CEL) and Long Short Distance Attention (LSDA) to model cross-scale interactions. CEL has the capability to merge multi-scale features with various kernel sizes, while LSDA enables the self-attention to capture feature interactions of both short distance and long distance. Experiments demonstrate that the proposed CrossFormer achieves performance improvement on image classification, object detection, instance segmentation and semantic segmentation tasks.

**Summary Of The Review:**

This paper proposed CrossFormer that aims to model cross-scale information through Cross-scale Embedding Layer (CEL) and Long Short Distance Attention (LSDA). The design of both modules is reasonable, and it demonstrates performance improvement on dense prediction tasks. However, due the potential limits of LSDA and DPB and missing comparison with some prior works, I would like to give a "marginally above the acceptance threshold" rating.

---

> ### Author Response · Authors · 2021-11-19
> **Comparison with the related work**
>
> ### Responses:
>
> 1. **Comparison with CoaT**: Both CoaT and CrossFormer introduce information of different scales, but there are some differences between them:
>
>    1. CoaT gets multi-scale features by mixing features from different layers, while we achieve multi-scale features at the same layer by using convolutions of different kernel sizes.
>    1. CoaT only uses multi-scale features at later layers of models. In contrast, we introduce multi-scale features at the start of each stage, thus tokens embeddings at all layers (not only those at later layers) could be seen as cross-scale.
>
>    We also compare CoaT and CrossFormer experimentally (FLOPs in an ascending order):
>
>    | Models | Params  | FLOPs | Accuracy |
>    | ---- | ----: | ----: | ----: |
>    | CoaT-LiTe Mini | 11M | 2.0G | 79.1% |
>    | **CrossFormer-T** | 27.8M | 2.9G | **81.5%** |
>    | CoaT Mini | 10M | 6.8G | 81.0% |
>    |                   |        |       |           |
>    | CoaT-Lite Small | 20M | 4.0G | 81.9% |
>    | **CrossFormer-S** | 30.7M | 4.9G | **82.5%** |
>    | CoaT Small | 22M | 12.6G | 82.1% |
>
>    Due to limited space, we have to sketch this discussion in Section 2, and detail it in the appendix C.1.

---

> > ### Comment · Reviewer_GtDH · 2021-11-30
> > **Thank you for your response!**
> >
> > The response has solved some of my questions. However, I still hold some concerns about the effectiveness and value of the dynamic position bias (DPB). Thus, I would like to maintain my weak accept rating.

---

> ### Author Response · Authors · 2021-11-19
> **Responses to the concerns and questions about our method**
>
> ### Glossary:
> - CEL: Cross-scale Embedding Layer proposed in our paper, which appears at the start of each stage and provides cross-scale embeddings for the model.
> - LSDA: Long Short Distance Attention proposed in our paper, which applies SDA and LDA alternately for short distance and long distance attention.
> - DPB: Dynamic Position Bias, a new position representation which not only supports dynamic image/group size, but also achieves similar accuracy to the popular relative position bias (RPB).
>
> ### Responses:
>
> 2. **The choice of $G$ and $I$ for LDA**: As said in Section 3.2.1, $G$ is fixed for SDA. While for LDA, $I$ is fixed regardless of image size (i.e., intervals for four stages are always [8, 4, 2, 1] whatever the image size is). Thus, enlarging image size will lead to a larger $G$ for LDA, and the elements in the group will not be scattered.
>
> 3. **DPB vs. RPB**
>
>    1. **Advantages of DPB over RPB**: The comparison between DPB and RPB is shown in the following table. The main contribution about DPB is **supporting dynamic group size**. In practice, image size of dense prediction tasks is often variable, and the group size for many vision transformers varies along the image size (e.g., PVT, CrossFormer, RegionViT, etc.). Therefore, compared with RPB that only supports fixed group size, DPB applies to all existing vision transformers, whether their group size is fixed or variable. When the group size is fixed, DBP is equivalent to RPB in the test phase, so it is reasonable that they achieve similar accuracies on ImageNet (where image/group size is fixed).
>       - Previous vision transformers (e.g., PVT) often resort to absolute position embedding (APE), and upsampling APE through linear interpolation when handling dynamic group size. Nevertheless, results in Table 7(b) of this paper show that RPB/DPB performs better than APE.
>
>     | Methods | Fixed Group Size (E.g., Swin Transformer) | Dynamic Group Size (E.g., PVT, CrossFormer, RegionViT, etc.) | Accuracy |
>     | - | - | -- | - |
>     | APE| Applicable| Applicable (through upsampling)| Baseline |
>     | RPB| Applicable| **NOT** applicable| Better   |
>     | DPB (ours) | Applicable, equivalent to RPB in the test phase (shown in the appendix A.2) | Applicable | Better   |
>
>    2.  **Examples of dynamic group size**: Models for detection/segmentation are good examples because they often require dynamic image/group size. We take two different input image sizes as examples below. We can see that the group size for LDA varies along the image size (this table takes the Stage-1 of a CrossFormer as an example, while the LDA's group size for other stages behaves in a similar way). RPB takes the relative position between two embeddings as index (e.g., $(\Delta x, \Delta y)=(1, 2)$) and draw the position bias from a fixed-size matrix (e.g., $B\in R^{15 \times 10}, bias=B[1,2]$). A larger group size may indicate a larger index than the size of the matrix (e.g., $(\Delta x, \Delta y)=(20, 15)>(15, 10)$). So, RPB does not support dynamic group size. Instead, as shown in Figure 3(b) and appendix A.2 of this paper, DPB can handle dynamic group size as a larger index (i.e., $(\Delta x, \Delta y)$) only affect the input range of DPB.
>
>    | Image Size | #Embeddings | SDA's Group Size (fixed) | LDA's Interval (fixed) | LDA's Group Size (dynamic) |
>    | - | - | - | - | - |
>    | 1280x960   | 320x240     | 14x14| 16x16 | 20x15|
>    | 960x640    | 240x160     | 14x14| 16x16| 15x10|
>
> 4. **Performance gain of CEL/LSDA + other vision transformers**
>
>    1. **CEL+Swin/NesT**: We implant our proposed CEL into NesT/Swin Transformer and find that CEL+NesT/Swin Transformer can further boost their performance. The specific results are shown below, which indicate that our proposed CEL is independent of previous work and acutally extending them. We did not do experiments with RegionViT because its code has not been released, and the relevant model cannot be reproduced perfectly. (The results have been supplemented in the appendix).
>
>    | Models       | Params | FLOPs | Accuracy |
>    | --- | ------: | -----: | --------: |
>    | Swin-T       | 29.0M  | 4.5G  | 81.3%     |
>    | Swin-T + CEL | 29.2M  | 4.8G  | **81.9%** |
>    | NesT-S       | 38.4M  | 10.4G | 83.3%     |
>    | NesT-S + CEL | 38.5M  | 10.6G | **83.8%** |
>    2. **LSDA+other vision transformers cannot be implemented**: LSDA is a substitute (rather than a supplement) of other vision transformers' attention mechanisms. We show the advantages of LSDA over the others in Table 7(a) of this paper, which shows that LSDA achieves 1.2%/0.6% absolutely higher accuracy than those attention mechanisms used by PVT and Swin.
>
> [1] Co-Scale Conv-Attentional Image Transformers, Xu et al, ICCV 2021
>
> [2] Pyramid Vision Transformer: A Versatile Backbone for Dense Prediction without Convolutions. ICCV 2021

---

### Author Response · Authors · 2021-11-19
**To all reviewers**

We thank all reviewers for their valuable comments. We are encouraged that the reviewers think our paper or our proposed method is simple (#R3/#R4), elegant (#R4), reasonable (#R1), sound/solid (#R2), neat (#R2), effective (#R2/#R3), clear (#R3), and easy to follow (#R1/#R3/#R4). Also, they think our experiments are thorough/comprehensive (#R2/#R3), strong (#R4), and achieve good/SOTA performance (#R1/#R2/#R3). While there are still some concerns or misunderstandings, we will try our best to resolve them for each reviewer.

---

### Decision · Program_Chairs · 2022-01-20

**Decision:**

Accept (Poster)

**Comment:**

The paper proposes several modifications to vision transformers: multiscale features, a variant of factorized attention, and "dynamic position bias". The proposed architecture with these modifications achieves strong results on classification, detection, and segmentation.

After considering the authors' responses, all reviewers are positive about the paper (reviewer K7wS mentioned upgrading to weak accept, but apparently forgot to do so). Main pros include clean architecture and strong empirical results. The main con is the somewhat limited novelty.

Overall, I recommend acceptance. While each of the proposed modifications might not be that unique, they are reasonably new in the context of transformers and their combination makes for a clean architecture that performs very well in practice.